# Autologous Stem Cell Transplantation in Hodgkin Lymphoma—Latest Advances in the Era of Novel Therapies

**DOI:** 10.3390/cancers14071738

**Published:** 2022-03-29

**Authors:** Yazeed Samara, Matthew Mei

**Affiliations:** 1Division of Hematology and Medical Oncology, Harbor-UCLA Medical Center, Torrance, CA 90502, USA; ysamara@dhs.lacounty.gov; 2Department of Hematology and Hematopoietic Cell Transplantation, City of Hope, Duarte, CA 91010, USA

**Keywords:** Hodgkin lymphoma, autologous stem cell transplantation, brentuximab vedotin, nivolumab, pembrolizumab

## Abstract

**Simple Summary:**

Standard second-line therapy for fit patients with relapsed and/or refractory Hodgkin lymphoma has long consisted of salvage therapy followed by autologous stem cell transplantation in responding patients. The introduction of the novel agents brentuximab vedotin, nivolumab, and pembrolizumab has transformed the management of Hodgkin lymphoma in multiple settings, and we review here the latest therapeutic advances in the management of the transplant-eligible patients.

**Abstract:**

Standard treatment for relapsed and/or refractory (*r*/*r*) Hodgkin lymphoma (HL) consists of salvage therapy, historically consisting of multiagent cytotoxic chemotherapy, followed by autologous stem cell transplantation (autoHCT) in responding patients. With this approach, most patients can proceed to autoHCT, of whom approximately half are cured. However, the introduction of the novel agents brentuximab vedotin (BV) and the checkpoint inhibitors (CPI) nivolumab and pembrolizumab has changed the decision making and peri-transplant decision making, as early incorporation of one or more of these agents can reduce or even eliminate the need for cytotoxic chemotherapy prior to autoHCT. Furthermore, post-autoHCT maintenance therapy with BV has also been shown to decrease relapse in high-risk rel/ref HL patients. In this review, we survey the current data regarding autoHCT in HL with a focus on pre-autoHCT salvage as well as maintenance strategies, and we also talk about the emerging data challenging the long-held dogma of chemosensitivity being a requirement for successful autoHCT.

## 1. Introduction

Hodgkin lymphoma (HL) is a B-cell lymphoma characterized by the presence of Reed–Sternberg cells surrounded by an inflammatory infiltrate. Overall, the prognosis is excellent, as 90% of patients with early stage disease and 70–80% of patients with advanced stage disease will be cured by frontline treatment. Unfortunately, up to 30% of patients with HL will have relapsed/refractory (*r/r*) disease, defined as not achieving complete remission (CR) with frontline therapy or disease relapse after prior CR [1,2,3]. Autologous stem cell transplantation (autoHCT) following second-line (2L) salvage therapy has long been the standard of care in this setting and cures about 50% of rel/ref patients [4,5]. Patients who achieve a complete metabolic remission (CR) as assessed by a PET/CT scan with 2L therapy prior to autoHCT have markedly superior long-term survival than those who do not [6,7,8].

The introduction of the CD30-directed antibody–drug conjugate brentuximab vedotin (BV) as well as the checkpoint inhibitors (CPI) nivolumab (nivo) and pembrolizumab (pembro), which are directed against the programmed death-1 (PD-1) receptor, have transformed the management of *r/r* HL, as all three agents demonstrate high single-agent activity in heavily pretreated disease [9,10,11]. Furthermore, BV given as post-autoHCT consolidation prolongs progression-free survival post autoHCT in patients with high-risk *r/r* HL [12]. Given the strong clinical activity of these agents, there has been ongoing interest in exploring their role in the second-line setting and especially in the context of autoHCT, either as pre-autoHCT salvage, post-autoHCT consolidation, or both. While 2L treatment has historically consisted of cytotoxic chemotherapy, BV monotherapy [13], BV with concurrent chemotherapy [14,15], BV with nivo [16], nivo monotherapy [17], and nivo/pembro + chemotherapy [17,18,19] have all been tested as 2L treatments and used successfully to bridge patients to curative intent autoHCT. In fact, the long-standing dogma that chemosensitivity is an absolute requirement prior to autoHCT has been challenged given the success of CPI-based salvage followed by autoHCT, even in previously chemorefractory patients [20].

In this review, we will discuss the current clinical data regarding salvage 2L therapy and post-autoHCT consolidation with a focus on the role of the novel agents BV and CPI as well as provide practice recommendations from our own practice. Of note, this review focuses on classical Hodgkin lymphoma only, as nodular lymphocyte-predominant HL is treated in a different manner and considered a separate disease [21].

## 2. What Is the Optimal Second-Line Therapy?

The choice of 2L therapy has become an increasingly crowded space, and at present there is no consensus as to the optimal therapy. In this section, we will discuss 2L options stratified broadly by mechanism (cytotoxic chemotherapy, BV-based, and CPI-based) as well as the emerging data supporting the concept that chemosensitivity may not be an absolute requirement for successful autoHCT.

### 2.1. Cytotoxic Chemotherapy

2L therapy in HL has long consisted of cytotoxic chemotherapy, and responding (i.e., “chemosensitive”) patients were typically consolidated further with autoHCT, which had been found superior to conventional dose chemotherapy alone [4]. Although multiple risk factors have been identified that are predictive for increased risk of post-autoHCT relapse [5], disease status immediately prior to autoHCT, as assessed by a PET/CT scan, has emerged as the most important one, with patients in complete remission (CR) faring far better than those in partial remission (PR), and patients who are refractory to salvage doing very poorly [6,7,8].

Multiple chemotherapy regimens have been tested in the 2L setting, all of which have roughly similar results, with most patients responding well enough to permit autoHCT [22,23,24,25,26]. Inter-regimen comparison has been difficult given that these data have largely been with smaller, single-arm trials, and no regimen has been shown to be superior in a prospective, randomized trial. Furthermore, many of these trials were conducted at a time response assessment by PET/CT scan, which further complicates the estimation of efficacy. The 2L salvage regimens with accompanying outcome data are listed in Table 1.

### 2.2. BV-Based Salvage Therapy

BV is an antibody-drug conjugate that consists of a chimeric anti-CD30 antibody conjugated to the microtubule-disrupting agent monomethyl auristatin E (MMAE) by an enzyme-cleavable linker. It is a highly active as a single agent in *r/r* HL, with a 75% ORR and 35% CR rates; a few patients achieving CR experience durable disease control [9,27].

#### 2.2.1. BV Monotherapy

Given the striking efficacy of BV, a few studies have been conducted evaluating its role as first salvage prior to autoHCT. A phase II study evaluating four doses of BV monotherapy as 2L therapy with disease assessment by PET/CT after the second and fourth doses was conducted. The overall response rate (ORR) was 75%, and 43% of patients achieved a CR (defined as a Deauville score of 1–3). In total, 50% of patients proceeded to autoHCT immediately post BV without intervening chemotherapy, while 38% of the patients eventually proceeded to autoHCT after further salvage treatment (88% total autoHCT rate). Of patients who proceeded to autoHCT, the 2-year PFS and OS were 67% and 96%, and for patients who proceeded directly to autoHCT after receiving BV alone, the 2-year PFS was 77% [13].

BV was also investigated as part of a planned sequential approach with augmented ifosfamide, carboplatin, and etoposide (ICE) in a separate phase II study. In total, 65 patients were treated with BV 1.2 mg/kg given on days 1, 8, and 15 for two 28-day cycles followed by a PET/CT scan. Patients in CR (defined as a Deauville score of 1–2) proceeded directly to autoHCT, while the other patients were treated with two cycles of augmented ICE. A total of 27% of patients had a CR post two cycles of BV, while 69% of patients receiving augmented ICE subsequently achieved CR; 76% of patients overall achieved CR with this approach. For all patients treated with autoHCT, the 3-year OS and PFS were 95% and 82%, respectively [28].

#### 2.2.2. BV Combination Therapy

Multiple regimens combining BV with cytotoxic chemotherapy have been studied. The safety and efficacy of 2L BV with bendamustine was evaluated in a phase I/II study of 55 patients with *r/r* HL, over half of whom had primary refractory disease. Patients received BV 1.8 mg/kg on day 1 and bendamustine 90 mg/m^2^ on days 1 and 2 of 3-week cycles for up to 6 cycles; patients could proceed to autoHCT after two cycles. After a median of two cycles, the ORR was 93% with 73.6% of patients achieving CR; 87% of the patients who attained CR did so after 2 cycles [14]. At a median follow up of 44.5 months, the 3-year OS and PFS were 92% and 60.3%, respectively (3-year PFS 67%.1% in patients who underwent autoHCT) [29].

Combination BV with ICE has also been tested in a phase I/II dose escalation trial of 45 patients who received dose-dense BV given on days 1 and 8 every 21 days together with ICE. Like the BV-bendamustine trial, the cohort predominantly consisted of patients at high risk of relapse: 64% of patients had primary refractory disease, and 24% had extranodal disease at relapse. The maximal tolerated dose of BV was 1.5 mg/kg on days 1 and 8, and the ORR was 94% with 74% CR. In total, 86% of patients subsequently proceeded to autoHCT, and the 2-year OS and PFS in the entire cohort were 97.8 and 80.4%, respectively. Toxicity was comparable to that seen with ICE alone, except for a 7% incidence of grade 2 or worse peripheral neuropathy and transaminase elevation [15]. 

Two other 2L BV-based salvage regimens have been studied in Europe. A phase I/II dose escalation trial of BV with ESHAP (etoposide, methylprednisolone, cytarabine, and cisplatin) was conducted by the Spanish Lymphoma Group (GELTAMO). In total, 66 patients were analyzed, of whom 61% had primary refractory disease. ORR was 91% with 70% CR; and the 30-month OS and PFS were 91% and 71%, respectively [30]. Another phase I/II investigated BV in combination with dexamethasone, cisplatin, and cytarabine (BV-DHAP) for patients with *r/r* HL. In total, 67 patients were enrolled and treated with three cycles of therapy over 21 days, of whom 43% had primary refractory disease and 29% relapsed within one year. PET/CT was performed after three cycles with those attaining at least a PR progressing to autoHCT. A total of 89% of patients completed all three cycles and 85% ultimately proceed to ASCT. The ORR and CR were 90% and 81%, respectively, with 2-year PFS and OS of 75% and 95% [32].

### 2.3. BV and Nivolumab

Given the non-overlapping toxicities of BV and nivo, the two were evaluated together as a chemotherapy-free 2L therapy in a multicenter, single-arm, phase I/II trial. In total, 93 patients were enrolled, of whom 91 received the full study treatment with a median follow up of 34.3 months. A total of 42% of patients had primary refractory disease, and an additional 30% of patients had experienced disease relapse within 1 year of completion of frontline treatment. The ORR for all treated patients was 85%, with 67% of patients achieving a CR, and the 3-year PFS for patients who proceeded to autoHCT was 91%. Notably, patients with primary refractory disease fared significantly worse, with a 3-year PFS of 61% vs. 90% in patients with relapsed disease. While a high rate of infusion reactions (44%) was observed, the treatment was well tolerated overall, and this trial provided the first proof of concept of the feasibility of chemotherapy-free salvage as a bridge to curative autoHCT [16]. 

### 2.4. CPI-Based Salvage

Both nivo and pembro are associated with high response rates as monotherapy in *r/r* HL [31,33]; also, for patients who are naïve to CPI and who are either naïve or responsive to BV, treatment with CPI is associated with longer PFS than BV [34]. Furthermore, with the increasing use of BV-based frontline therapy after the FDA approval of BV together with adriamycin, vinblastine, and dacarbazine for initial treatment of advanced stage HL [35], interest has grown in evaluating the role of CPI-based salvage independent of BV. At present, three trials of separate combinations reported preliminary results, all with CR rates well in excess of what have been reported with other salvage approaches, making checkpoint inhibition-based salvage a very promising avenue for further study.

Pembro with gemcitabine, vinorelbine, and liposomal doxorubicin (pembro-GVD) was studied as 2L therapy in 39 patients, of whom 49% had primary refractory disease and 38% had relapsed disease within 1 year of frontline therapy. Patients who achieved CR after 2–4 cycles proceeded to autoHCT. The ORR was an unprecedented 100% with 95% of patients achieving CR (92% after 2 cycles), and 95% of patients proceeded to autoHCT. Similar efficacy was noted across risk groups, and all transplanted patients remained in remission at median post-transplant follow up of 12.5 months. Notably, 68% of patients experienced engraftment syndrome at a median of 10 days, significantly exceeding historical control [18].

Preliminary results of pembro with ICE were presented at the American Society of Hematology (ASH) meeting in 2021: 42 patients were enrolled, and 37 patients were evaluable for efficacy, for whom the CR rate was 86.5% and the 24-month PFS was 88.2%. Notably, one patient died of cardiac arrest during stem cell collection, and another patient succumbed to acute respiratory failure early post-autoHCT, attributed to engraftment syndrome [19].

Finally, nivo in combination with ICE was also studied in a prospective, multicenter trial as first salvage treatment in *r/r* HL. In total, 39 patients were enrolled in the first cohort, and treatment consisted of nivo 3 mg/kg every 2 weeks for up to 6 cycles. PET/CT was performed after cycles 3 and 6, and patients who had not achieved a CR after cycle 6 went on to receive two cycles of nivo + ICE (NICE). After six cycles of single agent nivo, the ORR was 90%, with 77% of patients achieving a CR and 13% of patients a PR. Thus, most of the patients were able to proceed to autoHCT with nivo alone. The remaining seven patients received NICE, and 100% responded, with six patients achieving a CR (86%) and 1 PR (14%); 33 patients proceeded to autoHCT directly after protocol therapy, and the 2-year post-autoHCT OS and PFS were 97% and 94%, respectively [36].

As far as real-world data are concerned, an important multicenter analysis was conducted of 78 patients who underwent autoHCT after prior CPI. All of these patients had previously had insufficient responses to proceed to autoHCT after at least two prior lines of systemic therapy, and intervening therapy post-CPI was allowed. The median number of prior lines of therapy before CPI was 3, and 71% of patients were refractory to the last line of pre-CPI treatment; overall, 60% of patients were refractory to two or more salvage regimens. A total of 26% of patients received further salvage pre-autoHCT after CPI. The 18-month PFS for the entire cohort was 81% and was 78% in patients who were refractory to 1 or 2 lines of therapy immediately pre-CPI. Responsiveness to CPI and not chemotherapy was the key predictive factor for post-autoHCT outcomes [20]. This effect was hypothesized to be the result of CPI sensitizing the HL to subsequent chemotherapy, as has been reported elsewhere [37].

### 2.5. Can autoHCT Be Omitted?

Given the success of modern salvage therapies, the question has been raised in multiple quarters regarding whether or not autoHCT can be omitted in patients who achieve a CR. We do not advise this approach outside of a clinical trial for a number of reasons. While BV monotherapy and CPI therapy can both result in durable remissions, only a small minority of patients will sustain a response for years with either. For instance, 5-year follow-up of BV monotherapy found that 9% of patients achieved a long-term remission exceeding 5 years with no further therapy [27]. In the case of the CPI, both nivo and pembro were found to have 5-year PFS under 20% [38,39]; of the four patients who achieved CR with nivo in the nivo + ICE trial who refused autoHCT, three had relapsed [36]. 

As far as BV with nivo, only five patients did not proceed to autoHCT in the combination salvage trial from Advani et al., of whom two had progressive disease while on therapy and two others only achieved PR as their best response. The long-term outcomes of those patients are not provided in the manuscript. Similarly, there are no data for the long-term results of patients who receive cytotoxic chemotherapy together with CPI who then forgo autoHCT; therefore, whether this approach could be potentially curative is unknown. Nonetheless, given the very low rates of morbidity and mortality associated with autoHCT with modern supportive care, as well as the known curative potential of this therapy, we still recommend proceeding to autoHCT in all eligible *r*/*r* HL patients who respond well to salvage therapy. A trial of BV with nivo for patients who are ineligible for or who decline autoHCT is ongoing, which will help assess the durability of this approach without autoHCT (NCT04561206).

### 2.6. Summary

At present, there are insufficient data to support a specific 2L regimen, and the ultimate choice of therapy is at the discretion of the treating physician and should be based on patient characteristics, prior therapy (for instance, patients who received BV in the 1L setting should probably receive non-BV 2L regimen), and physician comfort level. At present, it is not known whether the choice of 2L therapy affects post-autoHCT outcomes independent of disease status. Increasing use of BV in the frontline setting has led to an interest in CPI-based 2L treatment, and results of at least three separate combinations of CPI with chemotherapy have been reported, all of which show CR rates exceeding historical controls although concerns regarding engraftment syndrome have been raised. We do not recommend omission of autoHCT for patients who achieve a CR to a salvage regimen including CPI and/or BV. Finally, patients who are refractory to cytotoxic chemotherapy but who respond well to CPI should still be considered for autoHCT, as sensitivity to chemotherapy should no longer be considered an absolute requirement for successful autoHCT.

## 3. Conditioning Regimens

The most widely used conditioning regimen in HL is BEAM (carmustine, etoposide, cytarabine, and melphalan). A large retrospective analysis was conducted by the Center for International Blood and Marrow Transplant Research (CIBMTR), examining results of lymphoma patients who underwent autoHCT between 1995 to 2008. In total, 1012 patients with HL were included, and BEAM was associated with lower mortality compared to other regimens in HL whereas total body irradiation (TBI)-based regimens were associated with an increased risk of relapse/progression [40].

Carmustine is known to cause pulmonary toxicity, and BEAM may result increased non-relapse mortality in patients with a poor baseline pulmonary reserve [41], which is a concern given the widespread use of bleomycin in the frontline setting. However, data regarding alternative conditioning regimens in HL are limited. Bendamustine with etoposide, cytarabine, and melphalan (BeEAM) has been evaluated in several smaller single-center studies, primarily in non-Hodgkin lymphoma. For instance, a single-center, retrospective analysis of 41 patients conditioned with BeEAM followed by autoHCT in Canada between 2015 to 2019 included only seven patients with HL. In total, 32% of patients experienced nephrotoxicity, with the majority being grade 1–2 (one patient experienced grade 3 nephrotoxicity), a finding that had been reported elsewhere as well [42]. One sudden death was observed in the BeEAM cohort from cardiac arrest in a patient with a history of cardiac disease. Efficacy appeared comparable to a historical cohort treated with BEAM from the same institution with respect to 3-year OS (71% vs. 78%) and 3-year PFS (74% vs. 71%). 

A multi-center phase II SWOG trial also investigated a tandem autoHCT approach for *r/r* HL in 98 patients; 89 patients were ultimately treated with 82 undergoing tandem autoHCT. The first autoHCT was conditioned by high-dose melphalan, 150 mg/m^2^, and patients achieving at least stable disease then received a second autoHCT conditioned with either TBI, etoposide, and cyclophosphamide or carmustine, etoposide, and cyclophosphamide. The median follow-up was 6.2 years, with the 5-year OS and PFS being 91% and 55%, respectively; there were no treatment-related deaths within the first year of autoHCT although three patients developed therapy-related myelodysplasia [43].

Finally, the addition of anti-CD25 radioimmunotherapy (90-Yttrium conjugated to basiliximab) to the standard BEAM regimen (aTAC-BEAM) was tested in HL with no dose-limiting toxicity seen. In total, 25 patients, all of whom would have met the AETHERA inclusion criteria, were enrolled to the study. The estimated 2- and 5-year PFS and OS were 68% and 95%, respectively, and at 5 years the NRM was 0%, while seven patients (32%) relapsed after a median of 3.7 months [44]. A phase II study of this regimen is ongoing (NCT04871607).

Overall, with the expanding armamentarium of novel salvage and frontline regimens, it is difficult to ascertain whether alternative conditioning regimens confer any benefit. At our institution (City of Hope), we prioritize any open clinical trials; off protocol we consider BeEAM in patients with prior bleomycin lung toxicity or impaired pulmonary function pre-autoHCT and primarily use BEAM for other patients.

## 4. Post-autoHCT Consolidation

### 4.1. BV Consolidation

Given the relatively high rate of post-autoHCT relapse, a phase III, double-blinded, placebo-controlled trial of consolidation BV, given 30–60 days post-autoHCT (1.8 mg/kg every 3 weeks for up to 16 doses), was conducted (AETHERA), which enrolled 329 patients. Patients had to have at least one high risk feature, defined as primary refractory disease, relapse within 12 months of initial therapy, or extranodal involvement at relapse. The cohort overall reflected a high-risk population: 60% had primary refractory disease, nearly half (46%) had two or more salvage regimens, and only 34% of patients had a negative PET/CT scan prior to autoHCT.

Patients who received BV had a significantly longer PFS than those who did not (median 42.9 months vs. 24.1 months), and on 5-year follow-up the PFS difference remained statistically different (59% versus 41%, respectively) [12]. Of note, patients with ≥2 adverse risk factors (primary refractory disease or relapse within 12 months of initial therapy, extranodal involvement at relapse, B symptoms at relapse, less than a CR at auto-HCT, or requiring two or more salvage therapies before auto-HCT) seemed to derive the most benefit from consolidation. Unsurprisingly, patients receiving BV had significantly more treatment-emergent peripheral neuropathy (67% vs. 19%, grade 3+ in 10%), leading to dose delay or discontinuation in over half of the patients. Neutropenia was also more common in BV-treated patients with grade 3+ neutropenia in 29% of patients, and the median number of BV doses given was 10.5 [45].

Although no trial has prospectively examined the role of BV consolidation in patients receiving frontline BV, consolidation could be considered in high-risk patients who remain BV sensitive after a short course of BV before auto-HCT (e.g., CR to BV-based salvage). The AMAHRELIS study was a real-world analysis of 115 patients with *r/r* HL who received at least two doses of BV consolidation post autoHCT, most of whom (70%) received BV-based salvage prior to autoHCT. A total of 43% of patients had primary refractory disease, and 51% of patients received more than one line of salvage therapy. Overall, the 2-year OS and PFS were 96.4% and 75.3%, respectively, with no difference seen in patients with or without previous BV exposure [46].

### 4.2. CPI-Based Consolidation

PD-1 blockade has also been evaluated as post auto-HCT consolidation/maintenance therapy with the aim of improving rates of durable remission while avoiding the development of peripheral neuropathy with repeated BV dosing. In a phase II study of 31 patients, 87% of which who met the AETHERA criteria, pembro 200 mg IV was given every 3 weeks for up to eight cycles as consolidation therapy starting within 60 days of autoHCT. In the 28 evaluable patients, the PFS was 82% at 18 months, and 43% experienced an immune-related adverse event (irAE), which is defined as an autoimmune toxicity attributable to the CP. Most irAE were grade 1 or 2; one grade 3 pneumonitis was observed. Among patients who met the AETHERA criteria, the 19-month PFS was 85%, and the 19-month OS was 100% [47].

Similarly, nivo consolidation following auto-HCT in patients with *r/r* HL is also being studied: 37 patients have been enrolled to date in a phase 2 study of patients with high-risk *r/r* HL defined similarly as in the AETHERA trial (primary refractory, relapse within 12 months, or extranodal disease at relapse) and treated with nivo 240 mg IV every 2 weeks for up to 6 months, starting between 45 and 180 days post autoHCT. In very limited follow-up (median 9.2 months), the 6-month PFS was 92.1%, and the incidence of grade 3 or higher toxicity was 14% [48].

Finally, consolidation BV with nivo every 21 days for 8 cycles post auto-HCT was also evaluated in a multicenter phase 2 trial. In total, 59 patients were enrolled, all of whom had at least one of the following high-risk features: primary refractory disease, relapse within 1 year of completing therapy, extranodal disease at relapse, B symptoms at relapse, or more than 1 salvage regimen prior to autoHCT. Furthermore, 49% of patients completed eight cycles of both agents, and 76% completed eight cycles of at least one of the two drugs. The 24-month OS and PFS were 98% and 92%, respectively. Four patients discontinued BV early due to peripheral neuropathy (grade 3 in 2, grade 2 in 2), and seven patients discontinued nivo early for irAE. Of note, the study allowed for prior treatment with BV and/or CPI [49].

### 4.3. Our Practice

Overall, we favor BV consolidation in patients with two or more high-risk features per the AETHERA criteria; in patients with only one feature, we discuss the risks and benefits carefully and tend to forgo it. We have a relatively low threshold for either dose reducing or stopping BV altogether in the event of treatment-emergent adverse effects, such as peripheral neuropathy. We would consider BV consolidation in patients who were previously treated with BV if they have multiple high-risk features and achieved CR to prior BV-based salvage; however, we did not typically give it to patients who are refractory to or who relapsed after prior BV.

## 5. Conclusions

Although autoHCT is a relatively old technique, significant progress has been made in optimizing 2L therapy in HL, with an eye towards improving the rates of autoHCT as well as minimizing post-autoHCT relapse. With respect to pre-autoHCT salvage, the highest CR rates reported to date have been seen with checkpoint inhibitors + chemotherapy, albeit in small, single-center studies, and a large cooperative group trial of salvage chemotherapy +/− CPI is planned to definitively address whether the combination is truly superior to chemotherapy alone. BV-based salvage is also very reasonable for patients who did not receive frontline BV, and BV/nivo is an excellent chemotherapy-free option although it may not fare as well for patients with primary refractory HL. Accumulating data strongly suggest that chemosensitivity prior to autoHCT is no longer an absolute requirement provided that the disease demonstrates responsiveness to CPI, likely due to chemotherapy sensitization. No single conditioning regimen has been shown to be superior to BEAM, and consolidation therapy with BV remains standard for patients with high-risk disease, as per the AETHERA trial.

In summary:Patients with relapsed disease who received neither BV nor a CPI upfront have several effective salvage options (chemotherapy, BV with nivo, BV-based combinations, or even CPI-based regimens), none of which has been shown to be superior to others in a prospective, randomized trial. Of note, BV/nivo may not be the best option for patients with primary refractory disease.Cytotoxic chemotherapy remains an acceptable 2L option with no single regimen clearly recommended over another.Patients with chemorefractory disease who respond well to a CPI should still be considered for autoHCT given the favorable results of autoHCT post CPI.Patients who receive pre-autoHCT CPI should be monitored in the early post-autoHCT setting for engraftment syndrome given the high rates seen with pembro-GVD and the two deaths seen with pembro and ICE.We favor BEAM conditioning for most patients off protocol with consideration of BeEAM in patients with bleomycin lung toxicity or significantly impaired lung function on pre-autoHCT assessment.We favor post-autoHCT BV consolidation in patients with at least two high-risk AETHERA criteria although we consider it for all patients meeting at least one of the high-risk criteria.

## Figures and Tables

**Table 1 cancers-14-01738-t001:** Salvage regimens.

Regimen	*n*	ORR (%)	CR (%)	Survival	Reference
Chemotherapy-based salvage					
ICE	65	88	26	43m OS 83%	[23]
				43m EFS 68%	
IGEV	91	81	54	3y OS 70%	[22]
				3y PFS 53%	
ESHAP	82	67	50	5y OS 73%	[25]
				5y PFS 78% (CR), 16% (PR)	
BEGEV	59	83	73	5y OS 78%	[24]
				5y PFS 59%	
BV-based salvage					
BV	56	75	43	2y post-HCT OS 93%	[13]
				2y post-HCT PFS 67%	
BV-ICE (sequential)	44	N/A	76	2y OS 95%	[27]
				2y EFS 80%	
BV-bendamustine	55	93	74	3y OS 92%	[28]
				3y PFS 60%	
BV-ICE (concurrent)	45	94	74	2y OS 98%	[15]
				2y PFS 80%	
BV-ESHAP	66	91	70	30m OS 91%	[29]
				30m PFS 71%	
BV-DHAP	67	90	81	2y OS 95%	[30]
				2-y PFS 75%	
CPI-based salvage					
BV-nivo	91	85	67	3y OS 93%	[16]
				3y PFS 77%	
Pembro-GVD	36	100	95	100% OS	[18]
				100% PFS	
Pembro-ICE	39	97	87%	2y PFS 88%	[19]
				27m OS 95%	
Nivo ± ICE	42	93	91	2y OS 95%	[31]
				2y PFS 72%	

*n* = number; ORR = overall response rate; CR = complete response; PR = partial response; ICE = ifosfamide, carboplatin, etoposide; IGEV = ifosfamide, gemcitabine, etoposide, vinorelbine; BEGEV = bendamustine, gemcitabine, vinorelbine; OS = overall survival; EFS = event-free survival; PFS = progression-free survival; BV = brentuximab vedotin; ESHAP = etoposide, methylprednisolone, cytarabine, cisplatin; DHAP = dexamethasone, cytarabine, cisplatin; Pembro = pembrolizumab; GVD = gemcitabine, vinorelbine, liposomal doxorubicin; Nivo = nivolumab.

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
