# Peer review of "Autologous Stem Cell Transplantation in Hodgkin Lymphoma—Latest Advances in the Era of Novel Therapies"

_cancers, 2022, doi:10.3390/cancers14071738_

Round 1
Reviewer 1 Report
This is a comprehensive, well written review on the latest, novel therapeutic options in relapse/refractory patients with Hodgkin's lymphoma, prior to autologous stem cell transplantation, focusing on 2nd line treatments, as well as consolidation options post-autologous stem cell transplantation in high risk patients.
The authors review the literature regarding traditional 2nd line chemotherapy salvage protocols and elaborate on current data on brentuximab vedotin (BV) monotherapy, BV-combined with chemotherapy, BV+checkpoint inhibitors, as well as checkpoint inhibitors, with or without chemotherapy, in the setting of 2nd line salvage in R/R Hodgkin lymphoma. They also review ASCT conditioning regimens for patients with Hodgkin's lymphoma and possible consolidation protocols for high risk patients post ASCT.
Minor revisions:
- Although there is only limited data, long term outcome of patients that achieved CR after BV/checkpoint inhibitors and did not proceed to ASCT should be mentioned.
- Considering the promising outcomes achieved with checkpoint inhibitors and BV+checkpoint inhibitors, including high CR rates, it would be interesting to add the authors perspective on the necessity/relevance of ASCT in patients who attain CR- relating to the very scares literature that exists, future studies and the authors personal opinion on the subject.
- Section 2.4 CPI based salvage: the authors relate to CPI+chemotherapy and don't mention CPI monotherapy. As they do mention the results of the BV monotherapy (Younes et al., JCO 2012) although most patients had ASCT prior to the treatment, the results of pivotal studies on CPIs monotherapy in the setting of R/R HL (Ansell, NEJM 2015, Armand, JCO 2018-CHECKMATE 205) and the comparison between BV and CPI in this setting (Kuruvilla et al., Lancet oncology 2021- KEYNOTE 204) should be mentioned in this review.
- Section 4.3 is now designated as "summary", yet it is more of the authors' personal practice. Please change this section headline.
- Abstract, 1st sentence- should be "Standard treatment FOR relapsed and/or refractory…"
- Introduction, line 4-5: "up to 30% of patients.." of patients is written twice in a row . Please omit.
- Introduction, last sentence- "with a focus both on the role of novel therapeutics (i.e….)." – novel therapeutics and?. Perhaps it should be rephrased "with a focus on the role of both novel agents (BV and CPI)"?
- Section 2.4, last paragraph: "The 18 months PFS…"- the sentence is too long, please rephrase. Perhaps "18 months PFS for the entire cohort was 81% and 78% in patients who were refractory... immediately pre-CPI. Responsiveness to CPI and not chemotherapy..."
- Conclusions: …the highest CR rates reported to date have been seen with checkpoint inhibitors+chemotherapy", Please add that this was in relatively small cohorts.
- Section "In Summary", part 1. – I believe that when the authors write BV/Nivo they mean BV+Nivo as a combination and not BV or Nivo, as now implicated. Please change.
Author Response
We appreciate the detailed comments which we believe have strengthened the manuscript. Please see our responses below:
- Data are limited on this point. We have amended the paper with a separate paragraph discussing this issue. Key points: BV monotherapy is associated with a 5-year PFS of < 10%, and both nivo/pembro have a 5-year PFS < 20% (CHECKMATE-205 and KEYNOTE-087 5-year updates). Almost all patients who received BV/nivo went to autoHCT although 5 did not of whom 2 were in PR and 2 had already progressed. There is an ongoing trial of 2L BV/nivo for patients who are ineligible for or who refuse transplant and this will help to assess durability of responses for BV/nivo.
2. We think that autoHCT is still necessary. There is not a clear survival plateau with long-term follow-up with CPI alone although there may be a few long-term cures (5-y PFS < 20% for both nivolumab/pembrolizumab in their most recent respective updates). 3/4 patients who received 2L nivolumab on the City of Hope trial who declined ASCT relapsed and required alternate therapy. There are very few BV/nivo patients who did not go to auto (mostly because of primary progression). However, there is a 2L trial looking at BV/nivo absent autoHCT which will hopefully provide some data on the durability of this combination. Finally, there are no data on chemo + PD-1 without autoHCT so whether this could be a potential avenue towards cure without autoHCT is unknown. Given the relatively low morbidity and very low mortality of autoHCT (and the fact that is a known quantity with a high cure rate), we find that the present data are insufficient to recommend omission of autoHCT except in the context of a well-designed trial.
3. We have added these studies in section 2.4.
4. Changed
5. Changed
6. Changed.
7. Changed (role of both novel agents).
8. Changed as recommended.
9. Changed as recommended.
10. Changed as recommended.
Reviewer 2 Report
This is a very-well written review about the role of autologous stem cell transplanatation in Hodgkin lymphoma. In my opinion it deserves consideration for publication in Cancers.
Only three minor mistakes/comments:
1. Page 4, 3rd paragraph: experienced (not exrpeicned)
2. Page 6, last paragraph, first sentence...
Finally, BV plus nivolumab (1.8 mg/kg of BV and 3 mg/kg nivolumab every 21 days for 8 cycles) post auto-HCT...
3. Page 7, first sentence define better irAE.
Author Response
We appreciate the reviewer's comments which we feel have helped us strengthen the paper.
- Changed
- Changed
- irAE is defined and clarified.
Reviewer 3 Report
This review is short but thorough description of existing second-line salvage therapy options prior to autologous stem cell transplantation in HL and posttransplantation consolidation options.
My only question is if these options are the same for all HL subtypes? It would be beneficial to add this information in discussion about different available options.
Author Response
We appreciate the reviewer comments. We have made explicit mention that this review only covers classical Hodgkin lymphoma and not nodular lymphocyte Hodgkin lymphoma